# Self-Assembly by Tridentate or Bidentate Ligand: Synthesis and Vapor Adsorption Properties of Cu(II), Zn(II), Hg(II) and Cd(II) Complexes Derived from a Bis(pyridylhydrazone) Compound

**DOI:** 10.3390/molecules26010109

**Published:** 2020-12-29

**Authors:** Hong-Juan Liu, Rui Yi, Dong-Mei Chen, Chao Huang, Bi-Xue Zhu

**Affiliations:** Key Laboratory of Macrocyclic and Supramolecular Chemistry of Guizhou Province, Guizhou University, Guiyang 550025, China; hjliu0323@163.com (H.-J.L.); lujhgzu@163.com (R.Y.); chendm3807@163.com (D.-M.C.)

**Keywords:** pyridylhydrazone, complex, crystal structure, vapor adsorption

## Abstract

Four complexes, [Cu_4_**L**_2_(OCH_3_)_2_(CH_3_OH)_2_]·2H_2_O (**1**), [Zn_2_**L**_2_Cl_4_]·2H_2_O·2CH_3_OH (**2**), [Hg_2_**L**_2_Br_4_]·4CH_3_OH (**3**), and {[Cd**L**_2_Cl_2_]·4H_2_O·4CH_3_OH}_n_ (**4**), have been synthesized and characterized from a bis(pyridylhydrazone) ligand (**L**) with copper(II), zinc(II), mercury(II) or cadmium(II), respectively. Complex **1** exists as a centrosymmetric tetranuclear dimer with **L** as deprotonated tridentate ligand. Complexes **2** and **3** exist as centrosymmetric metallamacrocycles with **L** as bidentate ligand. Complex **4** exists as a 1D looped-chain coordination polymer. The thermal stabilities and vapor adsorption properties of the four complexes were investigated as well.

## 1. Introduction

Within the field of supramolecular chemistry, hydrazone derivatives have been one of the most important classes of flexible and versatile polydentate ligands which show very high efficiency in chelating with transition metal ions [1,2,3,4,5,6]. The coordinating ability of hydrazones is possible due to the nucleophilic character of the nitrogen atoms of the triatomic structure C=N-N of the azomethine group [7,8]. So far, hydrazone-based metal complexes have received considerable attention from chemists in many applications such as chromogenic reagents in the spectrophotometric determination of transition-metal ions, metal extracts and biologically active compounds [9]. They have also been demonstrated to possess diverse pharmacological properties [10,11,12,13,14,15,16,17], catalytic properties [18,19], adsorption properties [20,21], electrochemical properties [22], luminescent properties [23,24], etc.

Pyridyl moiety is probably the most popular building block for the construction of metal-organic networks due to its strong coordination ability to metal ions [25,26,27]. In recent years, bis(pyridylhydrazone) ligands are often used in the construction of novel supramolecular architectures and promising candidates for metallo-anion receptors. Coordination polymers with organic ligands based on dipyridylamide moieties reveal that dipyridyl groups are a good choice for connecting metal centers and the hydrogen bonding of amide functionality can add extra dimensionality to the resulting structures. Several beautiful examples have been reported recently and further demonstrate the potential in the organization of the primary molecules in the solid state [28,29,30,31].

In this work, a bis(pyridylhydrazone) ligand (**L**) was synthesized by the Schiff base condensation reaction of 5-(tert-butyl)-2-hydroxyisophthalaldehyde with nicotinic hydrazide (Scheme 1). Four complexes have been synthesized and characterized from the bis(pyridylhydrazone) ligand with Cu(II), Zn(II), Hg(II), and Cd(II), respectively. The different geometries of the complexes disclose that the coordination sites play important roles in the formation of different structures.

## 2. Results and Discussion

### 2.1. Description of Crystal Structures

#### 2.1.1. Crystal Structure of Complex **1**

The reaction between the ligand and Cu(Ac)_2_ yielded green crystals of tetranuclear [Cu_4_**L**_2_(OCH_3_)_2_(CH_3_OH)_2_]·2H_2_O (**1**). Complex **1** crystallizes in the triclinic space group *P*ī. In the complex, the Addison *τ*_5_ parameters of copper(II) are 0.09 and 0 for Cu1 and Cu2 (*τ*_5_ = 0 for a perfectly square-pyramidal geometry, and *τ*_5_ = 1 for a perfectly trigonal-bipyramidal geometry) [32], suggesting that both copper atoms adopt a square-pyramidal geometry (Figure 1a). The structure can be considered as a centrosymmetric tetranuclear dimer complex which consists of two trianionic ligands, two methanol molecules, two methoxide ions, and four copper(II) ions. The copper atom is coordinated with three oxygen atoms and one nitrogen atom in the basal positions, and a methanol oxygen atom (for Cu1) or a methoxide oxygen atom (for Cu2) in the axial position. In the molecule, the Cu1-Cu2 distance is 2.957(4) Å, which is consistent with similar complexes found in the literature [33,34]. Noticeably, [Cu2(μ_3_-OCH_3_)_2_Cu2a] forms a square with Cu2-Cu2a separation of 2.973(5) Å and the distance of the two μ_3_-methoxide oxygen atoms bridged by Cu2/Cu2a is 2.969(4) Å, which is well comparable with previously reported copper complexes [35]. The packing view shows O–H···N hydrogen bonds between two neighboring ligands constitute an infinite 1D supramolecular chain (Figure 1b).

#### 2.1.2. Crystal Structure of Complexes **2** and **3**

The reaction of the ligand with ZnCl_2_ or HgBr_2_ in the MeOH–DMF solution gave the complex [Zn_2_**L**_2_Cl_4_]·2H_2_O·2CH_3_OH (**2**) or [Hg_2_**L**_2_Br_4_]·4CH_3_OH (**3**) as colorless block crystals. Both of the two complexes crystallize in triclinic space group *P*ī. Complexes **2** and **3** are isostructural except for the variation in the lattice solvent molecules, and they exist with very similar coordination configurations. Therefore, only the crystal structure of complex **2** is discussed herein. Complex **2** exists as a centrosymmetric 36-membered binuclear metallamacrocycle, which is composed of two Zn(II) ions, two ligands, four Cl anions, two methanol and two H_2_O molecules. Each Zn(II) is four-coordinated by two Cl anions and two pyridyl nitrogen atoms with a slightly distorted tetrahedral geometry deduced by its structural parameter *τ*_4_ = 0.91 (*τ*_4_ = 0 for a perfectly a square-planar geometry, and *τ*_4_ = 1 for a perfectly tetrahedral geometry) [36]. The Zn1–N1, Zn1–N6a, Zn1–Cl1 and Zn1–Cl2 bond lengths are 2.079(3), 2.060(3), 2.2264(16) and 2.2070(16) Å, respectively. The N–Zn–Cl bond angles are in the range of 104.46(8)°~108.21(9)°. The bond angles for Cl1–Zn–Cl2 and N1–Zn–N6a are 123.80(5)° and 104.21(12)°, respectively, which are consistent with those found in the similar complexes [37]. The dihedral angles between the pyridyl and the central benzene ring, and the neighboring pyridyl ring are 30.45° and 76.48°, respectively. The dinuclear structures connect to each other through N2–H2A···O3 hydrogen bond interactions to form 1D channels in the overall three dimensional network (Figure 2b).

#### 2.1.3. Crystal Structure of Complex **4**

The reaction of the ligand and CdCl_2_ in MeOH–DMF (4:1, *V*/*V*) gave the complex {[Cd**L**_2_Cl_2_]·4H_2_O·4CH_3_OH}*_n_* (**4**) as yellow block crystals. Complex **4** crystallizes in the triclinic space group *P*ī. The cadmium atom is six-coordinated by two chlorine atoms and four pyridyl nitrogen atoms from four adjacent ligands in a slightly distorted octahedral geometry. The four pyridyl nitrogen atoms constitute the basal plane of the octahedron, and the two chlorine atoms are located at the axial positions (Figure 3a). The ligands act as bidentate building blocks by pyridyl nitrogen atoms, linking the Cd(II) ions to form a 1D looped-chain of 36-membered macrocycles propagating along the *c* axis with the Cd···Cd distance of 16.68 Å. Each macrocycle is found to be associated with four methanol molecules via O-H···O, N-H···O or O-H···Cl hydrogen bonds (D···A = 2.726(10)–3.112(4) Å). Intermolecular hydrogen bonding interactions (N2-H2A···O3) and π···π stacking interactions between the adjacent benzene rings (the centroid-to-centroid distance is 3.912 Å) are formed to generate a 2D framework along the *bc* plane (Figure 3b), which is further linked by C22-H22···Cl1 hydrogen bond interactions, resulting in a 3D porous supramolecular network (Figure 3c).

#### 2.1.4. Structural Comparison

The organic ligands and metal ions play important roles in the formation of the final coordination frameworks and topologies. As described above, self-assembly of the ligands with Cu(II), Zn(II), Hg(II), and Cd(II) lead to 0D or 1D structures, which are all obtained in similar solvent evaporation conditions (MeOH/DMF). The structural analysis reveals that each copper(II) ion is five-coordinated with a square-pyramidal geometry. The zinc(II) and mercury(II) ions are four-coordinated with a slightly distorted tetrahedral geometry, and the cadmium ion is six-coordinated with a distorted octahedral geometry. Besides, it was found that the bis(pyridylhydrazone) compound acts as a tridentate deprotonated ligand in complex **1**, while as a bidentate ligand in complexes **2**–**4**. In short, the diversity of coordination preferences of metal ions as well as the nature of the ligands affect the assembly of the architectures.

### 2.2. TG Analyses of the Complexes

Thermal stabilities of the four complexes were measured by thermogravimetric analysis (TGA) between 25 and 800 °C in the N_2_ atmosphere at the heating rate of 10 °C min^−1^ (Figure 4). Complex **1** shows a slight weight loss from room temperature to 200 °C corresponding to the release of two H_2_O molecules (observed weight loss 2.7%, calculated 2.8%). The complex began to decompose from about 200 °C due to the release of the methanol molecules and methoxide ion, as well as the decomposition of the organic frameworks. For complex **2**, there is a weight loss (5.2%) in the range of 25 to 140 °C, which is attributed to the loss of solvent molecules (calculated 8.6%). The organic framework begins to decompose at above 400 °C. Complex **3** shows a continuous weight loss (9.6%) in the temperature range of 25 to 240 °C, which is due to the loss of methanol molecules (calculated 10.1%). The complex begins to decompose at above 250 °C with a sharp weight loss, which is accompanied by the elimination of HgBr_2_. Complex **4** exhibits a slight weight loss (6.6%) in the temperature range 25 to 290 °C, then begins to decompose the organic frameworks with a sharp weight loss. The results show that the solvent molecules can affect the thermal stability of the complexes.

### 2.3. Adsorption Measurements of the Complexes

Considering different hydrogen bond interactions and porous structures in the complexes, we explored methanol vapor adsorption for the four complexes at room temperature (Figure 5). The samples activation occurred at 353 K in a dynamic vacuum for 8 h until the outgas rate was less than 4 mmHg·min^−1^. Vapor adsorption isotherms were obtained by a Micromeritics ASAP 2020 system under the methanol vapor atmosphere. The Langmuir surface area and BET (Brunauer–Emmett–Teller) surface area for complexes **1**–**4** are listed in Table 1. As shown in Figure 5, the adsorption isotherms show a typical type III nature with the largest quantity adsorbed for complexes **1**–**4** are 7.81, 5.59, 6.36 and 6.72 mmol·g^−1^, which are similar to the complexes reported in previous references [38,39]. It is meaningful for the complexes to be potential materials as adsorbent of methanol vapor.

## 3. Experimental Section

### 3.1. Materials and Physical Measurements

All reagents and solvents were commercially available (Aladdin, Shanghai, China) and used without further purification. Melting points (uncorrected) were determined by an LTD 9100 apparatus (Electrothermal Engineering, London, UK). ^1^H NMR spectra were recorded in DMSO-*d*_6_ using a JNM-ECZ 400 MHz NMR spectrometer (JEOL, Tokyo, Japan). IR spectra were recorded on a VERTEX 70 FTIR instrument (Bruker, Germany) with KBr pellets in the range of 4000–400 cm^−1^ regions. Elemental analysis (C.H.N) were carried out on a Vario ELIII elemental analyzer (Elementar, Berlin, Germany). The crystal diffraction data were collected with a D8 VENTURE diffractometer (Bruker, Berlin, Germany). Thermogravimetric analysis (TGA) curves were carried out on a TGA-7 thermal analyzer (Perkin Elmer, Waltham, MA, USA) in the temperature region of 25–800 °C. The vapor adsorption measurements of methanol were performed by an ASAP 2020 (Micrometrics, Norcross, GA, USA) outfitted with a turbo molecular drag pump.

### 3.2. Synthesis

#### 3.2.1. Synthesis of the Ligand

5-(tert-butyl)-2-hydroxy-isophthalaldehyde was synthesized according to previous literature [40]. A solution of 5-(tert-butyl)-2-hydroxy-isophthalaldehyde (0.42 g, 2.0 mmol) in anhydrous ethanol (50 mL) was heated and stirred until dissolved, and nicotinic hydrazide (0.55 g, 4.0 mmol) in anhydrous ethanol (50 mL) was added dropwise. The mixture was refluxed with stirring for 8 h and then cooled to room temperature. The solvent was evaporated under reduced pressure to obtain 10 mL yellowish oily liquid, and 30 mL of dichloromethane was added to precipitate the yellowish solid, which on recrystallization from methanol gave a bright yellow solid. Yield: 0.76 g, 85.5%; mp: 252 ~ 253 °C. ^1^H NMR(DMSO-*d*_6_, 400 MHz): *δ* 1.34 (s, 9H, -CH_3_), 7.79 (s, 2H, Ar-H), 7.58 ~ 7.61 (m, 2H, Py-H), 8.28, 8.30 (d, *J* = 8.0 Hz, 2H, Py-H), 8.76 (s, 2H, Py-H), 8.78, 8.79 (d, *J* = 4.0 Hz, 2H, Py-H), 9.10 (s, 2H, CH=N), 12.34 (s, 1H, -OH), 12.34 (s, 2H, CONH). ESI-MS: m/z = 445.20 [M + H]^+^. Anal. Calcd. (%) for C_24_H_24_N_6_O_3_: C 64.85, H 5.44, N 18.91. Found (%): C 64.89, H 5.49, 18.84. FT-IR (KBr pellet, ν/cm^−1^): 3402 (m), 3180 (s), 2961 (s), 2862 (m), 1649 (s), 1617 (s), 1592 (s), 1562 (s), 1459 (m), 1418 (m), 1352 (s), 1304 (s), 1229 (m), 1198 (w), 1159 (s), 1123 (w), 1091 (w), 1066 (w), 1027 (w), 957 (w), 892 (m), 823 (w), 708 (s), 631 (w). 

#### 3.2.2. Synthesis of Complexes **1**–**4**

*[Cu_4_**L**_2_(OCH_3_)_2_(CH_3_OH)_2_]·2H_2_O**(**1**)*, Cu(Ac)_2_·H_2_O (39.9 mg, 0.2 mmol) in methanol (40 mL) was added dropwise with stirring to **L** (44.5 mg, 0.1 mmol) in DMF (5 mL) and the stirring was continued for 2 h. The solution was filtered off and left for slow evaporation at room temperature. Green crystals were obtained after a few days. The crystals were collected by filtration, washed with methanol, and dried to give complex **1** in a 43% yield. Anal. Calcd. (%) for C_52_H_60_Cu_4_N_12_O_12_: C 48.07, H 4.65, N 12.94; Found (%): C 48.18, H 4.53, N 13.06. FT-IR (KBr pellet, ν/cm^−1^): 3423 (s), 2957 (s), 1615 (s), 1586 (s), 1549 (m), 1505 (s), 1472 (m), 1411 (w), 1375 (s), 1312 (w), 1226 (m), 1194 (w), 1153 (m), 1080 (m), 1048 (m), 958 (w), 917 (w), 844 (w), 822 (w), 763 (w), 731 (m), 703 (m), 639 (m).

*[Zn_2_**L**_2_Cl_4_]·2H_2_O·2CH_3_OH (**2**)*, ZnCl_2_ (13.6 mg, 0.1 mmol) in methanol (40 mL) was added with stirring to **L** (44.5 mg, 0.1 mmol) in DMF (10 mL) and stirring was continued at room temperature for 3 h. Then the same process was used as for **1**. Yield (based on Zn): 38%. Anal. Calcd. (%) for C_50_H_60_Zn_2_Cl_4_N_12_O_10_: C 47.60, H 4.79, N 13.32; Found (%): C 47.52, H 4.85, N 13.26. FT-IR (KBr pellet, ν/cm^−1^): 3434 (m), 3192 (m), 2961 (s), 2865 (m), 1620 (s), 1553 (s), 1519 (s), 1475 (m), 1374 (s), 1316 (s), 1239 (m), 1201 (w), 1161 (m), 1099 (m), 963 (w), 909 (w), 836 (w), 778 (w), 733 (m), 703 (m), 651 (w).

*[Hg_2_**L**_2_Br_4_]·4CH_3_OH (**3**)*, HgBr_2_ (36.0 mg, 0.1 mmol) in methanol (40 mL) was added dropwise with stirring to **L** (44.5 mg, 0.1 mmol) in DMF (5 mL) and stirring was continued at room temperature for 3 h. Then the same process was used as for **1**. Yield (based on Hg): 32%. Anal. Calcd. (%) for C_52_H_64_Hg_2_Br_4_N_12_O_10_: C 35.94, H 3.71, N 9.67; Found (%): C 35.81, H 3.62, N 9.79. FT-IR (KBr pellet, ν/cm^−1^): 3542 (n), 3191 (s), 3025 (m), 2961 (s), 2861 (m), 1655 (s), 1616 (s), 1563 (s), 1460 (m), 1427 (m), 1379 (m), 1351 (s), 1295 (s), 1229 (w), 1197 (w), 1159 (m), 1123 (w), 1091 (w), 1051 (w), 957 (m), 895 (m), 826 (m), 756 (w), 702 (s), 638 (w).

*{[Cd**L**_2_Cl_2_]·4H_2_O·4CH_3_OH}_n_ (**4**)*, CdCl_2_·2.5H_2_O (22.8 mg, 0.1 mmol) in methanol (40 mL) was added dropwise with stirring to **L** (88.8 mg, 0.2 mmol) in DMF (10 mL). Then the same process was used as for **1**. Yield (based on Cd): 35%. Anal. Calcd. (%) for C_52_H_72_CdCl_2_N_12_O_14_: C 49.08, H 5.70, N 13.21; Found (%): C 49.18, H 5.60, N 13.31. FT-IR (KBr pellet, ν/cm^−1^): 3452 (m), 3202 (s), 3038 (s), 2957 (s), 2863 (w), 1659 (s), 1617 (s), 1559 (s), 1471 (m), 1428 (w), 1353 (s), 1289 (s), 1229 (w), 1200 (w), 1159 (m), 1093 (w), 1030 (m), 956 (w), 894 (m), 826 (w), 731 (w), 704 (s), 638 (w).

### 3.3. Crystallographic Data Collection and Structure Determination

Single-crystal X-ray data of complexes **1**–**4** were collected using a D8 VENTURE diffractometer (Bruker, Berlin, Germany). Intensities of reflections were measured using Mo Kα monochromatized radiation (λ = 0.71 073 Å). Data reductions and absorption corrections were performed by using the SAINT and SADABS programs implemented in the APEX2 software (version 1.2), respectively. The structures were solved by direct methods and refined by full-matrix least squares methods on *F*^2^ using the SHELXTL programs [41,42]. All non-H atoms were refined anisotropically. Hydrogen atoms were generated geometrically with fixed isotropic thermal parameters, and included in the structure factor calculations. Some solvent molecules in the four complexes are disorder and removed by Platon/Squeeze in the APEX2 software (version 1.2). Crystal data and structure refinement parameters are listed in Table 2. Selected bond lengths and angles for the complexes are listed in Appendix A. Hydrogen bonding lengths and angles are listed in Appendix A. CCDC reference numbers: 2041792 for **1**, 2041793 for **2**, 2041794 for **3**, and 2041795 for **4**.

## 4. Conclusions

In summary, we have synthesized and characterized four complexes derived from a bis(pyridylhydrazone) ligand with copper(II), zinc(II), mercury(II) and cadmium(II), respectively. The structural analysis reveals that each copper(II) ion is five-coordinated with a square-pyramidal geometry in the tetranuclear complex **1**. The zinc(II) and mercury(II) ions are four-coordinated with a slightly distorted tetrahedral geometry in the metallamacrocycle **2** or **3**. The cadmium ion is six-coordinated with a distorted octahedral geometry in the 1D looped-chain coordination polymer **4**. It is worth noting that the bis(pyridylhydrazone) compound acts as a tridentate ligand in complex **1**, while acting as a bidentate ligand in complexes **2**–**4**, which indicates that the nature of the ligands play an important role in the coordination networks. The adsorption measurements for the complexes show that they can be potential materials as the adsorbents of methanol vapor.

## Data Availability

The data presented in this study are available on request from the corresponding author.

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
