# Peer review of "Self-Assembly by Tridentate or Bidentate Ligand: Synthesis and Vapor Adsorption Properties of Cu(II), Zn(II), Hg(II) and Cd(II) Complexes Derived from a Bis(pyridylhydrazone) Compound"

_molecules, 2020, doi:10.3390/molecules26010109_

Round 1
Reviewer 1 Report
Review on molecules-1021445 (only crystallographic part)
The refinement procedure as such is not ok, based on very bad data and various checkcif alerts have not been addressed.
Regrettably, data for 1 and 2 have not been collected at low temperatures (273 K is not low) and even 223 K for 3 and 4 is no good standard.
Although data have been collected up to 2theta = 56-57° (1-3) or even 67° (for 4), data for refinement were cut at 2theta = 48-50 degree: just to please the R-values.
The results suffer from exceptionell large electron density residuals (2-5 !) as well as eventual split positions (check your .lst files !) that have neither been mentioned nor treated. Moreover, the reference “…some solvent molecules are disorder(d) and then removed by Platon (literature citation missing)….” (page 7, para 3.3) is not appropriate. Obviously no real attempts have been in order to refine reliable disorder models for all structures.
Although authors state that 2 and 3 are “isostructural” (i.e. isotypic) the two atom position sets are not consistent. So 2 (or 3) has to be re-refined accordingly and atom labelling for both structures should be the same, too. Additionally, the atom positions have to be shifted as the (geometric) centers of the molecules lie not within the unit cells.
Judging these gross refinement errors the manuscript is not acceptable for publication.
The discussion in para 2.1.1 to 2.1.3 , on the other hand, is pleasingly short and focused on the essential.
Some more points to be considered:
- For all derived geometric data throughout the text the relevant s.u.’s have to be given, e.g. page 2, para 2.1.1, line 61ff.
- All molecules 1-4 lie on special positions, this has to be mentioned in the discussion.
- All figures are much too small. Additionally, the 35% ellipsoids are just for cosmetic reasons and deny most of the electron density. 50% should be standard. Caption should read “Molecular structure of xx with anisotropic displacement ellipsoids drawn at the 50% probability level”. “level” – not “label” !!
- Para 3.3 and table S1 and S2: there are no “hydrogen bonding distances”, this would imply you measure the distance between two bonds. So it must read “hydrogen bond lengths”.
- Page 7, para 3.3, line 195: there are no “thermal factors”, it must read “All non-H atoms were refined anisotropically”
- Although H-atom treatment is sound the details (positions derived from geometric reasons and/or difference electron density maps as well as the refinement parameters as D-H, Uiso etc) must be given in para 3.3.
- Statements like “Single crystal X-ray diffraction analysis reveals that complex 2 crystallizes in the …..” (para 2.1.2, line 71ff) should be avoided. A short but precise “Both crystallize in triclinic space group P-1” is sufficient.
- Para 2.1.3, line 92 must read “…crystallizes in triclinic space group …” (delete “with”).
- Figure 1: caption must read “Molecular structures of 1 with anisotropic displacement ellipsoids drawn at 50% probability level”. “Thermal ellipsoids” is a very very outdated (and incorrect) expression.
Author Response
Review 1: The refinement procedure as such is not ok, based on very bad data and various checkcif alerts have not been addressed. Regrettably, data for 1 and 2 have not been collected at low temperatures (273 K is not low) and even 223 K for 3 and 4 is no good standard. Although data have been collected up to 2theta = 56-57° (1-3) or even 67° (for 4), data for refinement were cut at 2theta = 48-50 degree: just to please the R-values. The results suffer from exceptionell large electron density residuals (2-5 !) as well as eventual split positions (check your .lst files !) that have neither been mentioned nor treated. Moreover, the reference “…some solvent molecules are disorder(d) and then removed by Platon (literature citation missing)….” (page 7, para 3.3) is not appropriate. Obviously no real attempts have been in order to refine reliable disorder models for all structures. Although authors state that 2 and 3 are “isostructural” (i.e. isotypic) the two atom position sets are not consistent. So 2 (or 3) has to be re-refined accordingly and atom labelling for both structures should be the same, too. Additionally, the atom positions have to be shifted as the (geometric) centers of the molecules lie not within the unit cells. Judging these gross refinement errors the manuscript is not acceptable for publication. We are so sorry for the bad data of the crystal structures, since there are a few free solvent molecules in the four complexes, resulting in the various checkcif alerts. In order to please the R-values, we have to use Platon/Squeeze to reduce to the refinement errors. Some more points to be considered:
1.For all derived geometric data throughout the text the relevant s.u.’s have to be given, e.g. page 2, para 2.1.1, line 61ff. Done. The relevant s.u.’s have been given.
2.All molecules 1-4 lie on special positions, this has to be mentioned in the discussion. Done. We added some discussion in section “Structural Comparison”.
3.All figures are much too small. Additionally, the 35% ellipsoids are just for cosmetic reasons and deny most of the electron density. 50% should be standard. Caption should read “Molecular structure of xx with anisotropic displacement ellipsoids drawn at the 50% probability level”. “level” – not “label” !! Done.
4.Para 3.3 and table S1 and S2: there are no “hydrogen bonding distances”, this would imply you measure the distance between two bonds. So it must read “hydrogen bond lengths”. Done.
5.Page 7, para 3.3, line 195: there are no “thermal factors”, it must read “All non-H atoms were refined anisotropically”. Done.
6.Although H-atom treatment is sound the details (positions derived from geometric reasons and/or difference electron density maps as well as the refinement parameters as D-H, Uiso etc) must be given in para 3.3. Done, some details have been revised.
7.Statements like “Single crystal X-ray diffraction analysis reveals that complex 2 crystallizes in the …..” (para 2.1.2, line 71ff) should be avoided. A short but precise “Both crystallize in triclinic space group P-1” is sufficient. Done.
8.Para 2.1.3, line 92 must read “…crystallizes in triclinic space group …” (delete “with”). Done.
9.Figure 1: caption must read “Molecular structures of 1 with anisotropic displacement ellipsoids drawn at 50% probability level”. “Thermal ellipsoids” is a very very outdated (and incorrect) expression. Done. We have corrected the errors.
Reviewer 2 Report
In the paper authors describe the synthesis, structure, thermal and adsorption properties of four complexes of Cu(II), Zn(II), Hg(II) and Cd(II) ions with bis(pyridylhydrazone) Schiff base. Before I can recommend this manuscript to publication in Molecules, I need to get some clarification and supplementation.
Line 52: The formula of complex 1 given in “Results and Discussion” paragraph is inconsistent with that given in the abstract and experimental section.
Section 2.1.2.: Although compounds 2 and 3 are isostructural some information regarding complex 3 should be provided/discussed e.g. geometry around Hg(II) centre, comparison of both structure, information how different solvents affect the structure etc.
Section 2.2.: This part needs general improvement mainly with regard to the following points:
- Lines 114-117: “Complex 1 shows a slight weight loss from room temperature to 200 ℃ corresponding to the release of two H2O molecules (observed weight loss 2.7 %, calculated 2.8 %), as well as a major weight loss occurring at above 280 ℃ due to the decomposition of the organic frameworks.” What is happen with methanol and methoxide ion during heating (when there are lost?)?
- Lines 117-118: Why there is such a big difference in the calculated and found value attributed to loss of solvent molecules for complex 2?
- A more detailed description of the TG analysis for complexes 3 and 4 should be provided.
Author Response
Review 2: 1.Line 52: The formula of complex 1 given in “Results and Discussion” paragraph is inconsistent with that given in the abstract and experimental section. The description of complex 1 has been revised to “[Cu4L2(OCH3)2(CH3OH)2]·2H2O”.
2.Section 2.1.2.: Although compounds 2 and 3 are isostructural some information regarding complex 3 should be provided/discussed e.g. geometry around Hg(II) centre, comparison of both structure, information how different solvents affect the structure etc. Complexes 2 and 3 are isostructural except for the variation in the lattice solvent molecules, and they exist very similar coordination configurations. So we think it’s better to omit some details about the description of complex 2. Some discussions were added in the section.
3.Section 2.2.: This part needs general improvement mainly with regard to the following points: - Lines 114-117: “Complex 1 shows a slight weight loss from room temperature to 200 ℃ corresponding to the release of two H2O molecules (observed weight loss 2.7 %, calculated 2.8 %), as well as a major weight loss occurring at above 280 ℃ due to the decomposition of the organic frameworks.” What is happen with methanol and methoxide ion during heating (when there are lost?)? A detailed description of the TG analysis for complex 1 has been added. - Lines 117-118: Why there is such a big difference in the calculated and found value attributed to loss of solvent molecules for complex 2? Complex 2 is unstable at room temperature because of the VOC molecules (MeOH/H2O), which results in the weight loss at R.T. Thus the observed weight loss is lower than the calculated weight loss of weight. - A more detailed description of the TG analysis for complexes 3 and 4 should be provided. Done. More detailed description (observed and calculated weight loss) was provided.
Reviewer 3 Report
This paper describes the syntheses, crystal structures and methanol vapor adsorption of the four complexes with a bis(pyridylhydrazone) ligand. Findings of this paper is interesting. However, this paper contains a mistake and some insufficient descriptions. Thus, this paper is worth publishing in Molecules with minor revisions.
Some additional comments are listed below.
1) The cif data of complex 1 is wrong. H2 and H5 should be removed.
2) line 63, 82--83, 99--100: Descriptions of hydrogen bonds are insufficient. Authors should explain them more in detail. In addition, Figure 2b, and 3c should be revised because intermolecular interactions are difficult to understand in the present figures
3) line 103: Authors should explain what are weak interactions.
Author Response
Review 3:
- The cif data of complex 1is wrong. H2 and H5 should be removed.
Done. Structure of complex 1 was revised.
- line 63, 82--83, 99--100: Descriptions of hydrogen bonds are insufficient. Authors should explain them more in detail. In addition, Figure 2b, and 3c should be revised because intermolecular interactions are difficult to understand in the present figures.
A detailed description of hydrogen bonds were added in the corresponding part, Figure 2b, and 3c have been revised to improve the description of hydrogen bonds.
3.line 103: Authors should explain what are weak interactions.
The description of weak interactions has been revised to “C22-H22···Cl1 hydrogen bond interactions”.
Reviewer 4 Report
The submitted manuscript describes the synthesis of a new ligand and four transition metal complexes of it. The characterisation of the complexes is mainly through single crystal X-ray diffraction studies. Overall I feel that this short work is of good quality, well written with very nice figures, and that it is appropriate for the journal.
However there are some things which I would encourage the authors to attend to.
It would be good to include a brief description of the synthetic method for each of the complexes – this would, for example, help the reader understand why there are methanol molecules in the structure of 1.
There is inconsistency in the way that the structures are described. One includes bond metrics, one includes the space group, etc. Describing each in the same way, with the same information included, would be better. Also, it would be helpful to the reader to clearly state the ionisation state of the ligand in each case.
There are a few places where the compound numbers are not bolded – this should be checked.
The picture of compound 1 in Figure 1a show be bigger to make it easier to see.
I ask that the authors check the IR spectrum of compound 1 that is included. Given that the X-ray structure shows that, in this compound, the C=O groups are coordinated to the Cu ions, I might have expected there to be a shift in the position of the C=O band, but there is none?
And it would be good is there was some discussion about how the results of the TGA studies (and, to a lesser extent, the vapour adsorption studies) might relate to the structures of the compounds. In particular, if compounds 2 and 3 and so similar, why are their TGA traces so different?
Author Response
Review 4:
- It would be good to include a brief description of the synthetic method for each of the complexes – this would, for example, help the reader understand why there are methanol molecules in the structure of 1.
Some descriptions were added in the section of Structural Comparison.
- There is inconsistency in the way that the structures are described. One includes bond metrics, one includes the space group, etc. Describing each in the same way, with the same information included, would be better. Also, it would be helpful to the reader to clearly state the ionisation state of the ligand in each case.
We have described the structures of the complexes in the same way.
- There are a few places where the compound numbers are not bolded – this should be checked.
Done. We have checked the format carefully in the whole manuscript.
- The picture of compound 1in Figure 1a show be bigger to make it easier to see.
Done.
- I ask that the authors check the IR spectrum of compound 1that is included. Given that the X-ray structure shows that, in this compound, the C=O groups are coordinated to the Cu ions, I might have expected there to be a shift in the position of the C=O band, but there is none?
IR spectrum of compound was remeasured. As is shown in the figure below, a blue shift (from 1562 to 1505) can be observed in complex 1 compared to the ligand.
- And it would be good is there was some discussion about how the results of the TGA studies (and, to a lesser extent, the vapour adsorption studies) might relate to the structures of the compounds. In particular, if compounds 2and 3 and so similar, why are their TGA traces so different?
More discussion about TGA studies have been added in the revised-manuscript. The difference of TGA traces between complexes 2 and 3 can be attributed to properties of the ions (Zn and Hg). The final thermal product obtained for complex 2 was ZnO, while the thermal product of complex 3 can reduce to almost zero because of the elimination of HgBr2.
Round 2
Reviewer 1 Report
Review on molecules-1021445 Version 2
Sad to say, but all what has been demanded with the first review (see below) is still valid, as authors have done nothing!! to resolve the problems.
Just to run SQUEEZE on the data without any knowledge of the process nor knowing which and how many “unresolved” solvent molecules must be taken into account or trying to model disordered structure parts is absolutely unserious.
At least the SQUEEZE procedure requires that all squeezed parts of the structure must be included in the sum formula, the UNIT as well as the SFAC instructions accordingly.
Cutting these bad data sets at 2theta = 48° is good for nothing and not acceptable.
There are still wrong “bond distances” and “isotropic thermal parameters” in the text and supplement tables.
Authors seem not to understand the basics of a reliable structure refinement and are apparently incapable (or unwilling) to create a meaningful model of the solvent molecules.
This manuscript is not acceptable for publication.
First review:
The refinement procedure as such is not ok, based on very bad data and various checkcif alerts have not been addressed.
Regrettably, data for 1 and 2 have not been collected at low temperatures (273 K is not low) and even 223 K for 3 and 4 is no good standard.
Although data have been collected up to 2theta = 56-57° (1-3) or even 67°, data for refinement were cut at 2theta = 48-50 degree: just to please the R-values.
The results suffer from exceptionell large electron density residuals (2-5 !) as well as eventual split positions (check your .lst files !) that have neither been mentioned nor treated. Moreover, the reference “…some solvent molecules are disorder(d) and then removed by Platon (literature citation missing)….” (page 7, para 3.3) is not appropriate. Obviously no real attempts have been in order to refine a reliable disorder model.
Although authors state that 2 and 3 are “isostructural” (i.e. isotypic) the two atom position sets are not consistent. So 2 (or 3) has to be re-refined accordingly and atom labelling for both structures should be the same, too. Additionally, the atom positions have to be shifted as the (geometric) centers of the molecules lie not within the unit cells.
Judging these gross refinement errors the manuscript is not acceptable for publication.
The discussion in para 2.1.1 to 2.1.3 , on the other hand, is pleasingly short and focused on the essential.
Some more points to be considered:
- For all derived geometric data throughout the text the relevant s.u.’s have to be given, e.g. page 2, para 2.1.1, line 61ff.
- All molecules 1-4 lie on special positions, this has to be mentioned in the discussion.
- All figures are much too small. Additionally, the 35% ellipsoids are just for cosmetic reasons and deny most of the electron density. 50% should be standard. Caption should read “Molecular structure of xx with anisotropic displacement ellipsoids drawn at the 50% probability level”. “level” – not “label” !!
- Para 3.3 and table S1 and S2: there are no “hydrogen bonding distances”, this would imply you measure the distance between two bonds. So it must read “hydrogen bond lengths”.
- Page 7, para 3.3, line 195: there are no “thermal factors”, it must read “All non-H atoms were refined anisotropically”
- Although H-atom treatment is sound the details (positions derived from geometric reasons and/or difference electron density maps as well as the refinement parameters as D-H, Uiso etc) must be given in para 3.3.
- Statements like “Single crystal X-ray diffraction analysis reveals that complex 2 crystallizes in the …..” (para 2.1.2, line 71ff) should be avoided. A short but precise “Both crystallize in triclinic space group P-1” is sufficient.
- Para 2.1.3, line 92 must read “…crystallizes in triclinic space group …” (delete “with”).
- Figure 1: caption must read “Molecular structures of 1 with anisotropic displacement ellipsoids drawn at 50% probability level”. “Thermal ellipsoids” is a very very outdated (and incorrect) expression.
Reviewer 2 Report
The manuscript has been corrected. All my remarks were took into account.